# What is known from the existing literature about self-management of pessaries for pelvic organ prolapse? A scoping review protocol

Lucy Dwyer [1,2] Dawn Dowding,[1] R Kearney[2,3]

¹Division of Nursing, Midwifery and Social Work, The University of Manchester, Manchester, UK
²Saint Mary's Hospital, Manchester University NHS Foundation Trust, Manchester Academic Health Science Centre, Manchester, UK
³Institute of Human Development, Faculty of Medical & Human Sciences, University of Manchester, Manchester, UK

**Correspondence to**
Lucy Dwyer;
lucy.dwyer@mft.nhs.uk

## ABSTRACT

**Introduction** Pelvic organ prolapse (POP) can be managed with a pessary; however, regular follow-up may deter women from pessary management due to the inconvenience of frequent appointments, as well as preventing pessary users from autonomous decision-making. Pessary self-management, whereby the woman removes and inserts her own pessary may be a solution to these issues. However, there remains a number of uncertainties regarding the potential benefits and risks of pessary self-management. This scoping review aims to map available evidence about the subject of pessary self-management for POP to identify knowledge gaps providing the basis for future research.

**Methods and analysis** The scoping review will be conducted using the Joanna Briggs Institute scoping review methodology and reported in accordance with Preferred Reporting Items for Systematic Reviews and Meta-Analyses extension for Scoping Reviews guidelines. A search of Medline, CINAHL, Embase and PsycInfo will be undertaken to identify relevant articles which meet the eligibility criteria using the search terms 'pessary' and 'self-management' or 'self-care'. A hand search of the reference list of non-original research identified during the search but excluded, will be conducted for additional publications which meet the inclusion and exclusion criteria. Data relevant to the topic of pessary self-management will be extracted and critical appraisal of all included publications undertaken.

**Ethics and dissemination** No ethical or Health Research Authority approval is required to undertake the scoping review. However, it has been registered with The Open Science Framework (DOI 10.17605/OSF.IO/DNGCP). The findings will inform future research exploring pessary self-management and be disseminated via both a presentation at a national conference and publications in peer reviewed journals.

## Strengths and limitations of this study

► The scoping review will be undertaken following systematic and rigorous established guidelines to ensure transparency and reproducibility.
► Critical appraisal of the evidence will be undertaken to determine the weighting of included research findings, as well as identifying potential methodological strengths and limitations of the current evidence base.
► The identification and synthesis of data will be limited to published articles found on the Medline, CINAHL, Embase and PsycInfo databases and a hand search of reference lists.

or defecating and sexual dysfunction, all of which can significantly negatively impact a woman's quality of life.[1]

A pessary is a medical device which can be inserted into the vagina to provide mechanical support to the prolapsed organs. Pessary management offers women with prolapse comparable treatment outcomes to surgery in terms of reported symptoms and quality of life[2] and absence of bulge and no desire for further treatment.[3] This may be a particularly desirable option for women who have not completed their family, are unfit for surgical management or would simply prefer to avoid the risks that POP surgery entails.[4] There are a wide variety of pessaries available, offering a conservative, long-term management option to women. Pessary follow-up is required, tending to be 3–6 monthly.[5] At each appointment, the pessary is removed, the vaginal tissues examined and either a new pessary, or the same pessary replaced after cleaning. The need for regular follow-up is often cited as a reason why women opt for surgical management of POP due to the inconvenience of frequent appointments.[6–8] Moreover, clinician management means women are denied autonomy in how and when to use their

## INTRODUCTION

Pelvic organ prolapse (POP) is defined as the downward displacement of one or more of the pelvic organs including the uterus, vaginal compartments, bowel or bladder.[1] Common POP symptoms reported by women include seeing or feeling a vaginal bulge, a heaviness or a dragging sensation, difficulties voiding

BMJ

pessary. Furthermore, the cost of regular pessary follow-up appointments is not insignificant, with over 86 000 pessaries inserted annually in English NHS services alone.[9]

A solution to these frequently cited issues with pessary care may be pessary self-management. This entails the woman removing and reinserting her own pessary as required or desired. There remains a number of uncertainties regarding the benefits and possible risks of pessary self-management.[10] A preliminary search in May 2021 identified a recent review of the published evidence related to pessary self-management, which highlighted current uncertainties within the subject area.[10] These include whether pessary self-management improves a woman's satisfaction, quality of life and therefore long-term use of a pessary; whether pessary self-management is safe and what follow-up requirements are for women self-managing their pessary.[10] However, the review was limited to searching PubMed and had a focus of patient safety as indicated by the search terms used. Therefore, only five eligible studies were identified. A preliminary search of Medline, the Cochrane Database of Systematic Reviews and JBI Evidence Synthesis was conducted and no further systematic or scoping reviews on the topic were identified as being currently underway.

It is acknowledged that there is a lack of robust evidence regarding the risks and benefits of pessary self-management due to the small-scale, non-randomised nature of much of the evidence.[10] While there is currently a lack of robust evidence regarding the benefits of self-management, there is qualitative and observational evidence from women using pessaries about the benefits pessary self-management offers to them, including flexibility in how and when they use the pessary.[6 11]

This scoping review aims to map available evidence about the subject of pessary self-management for POP to identify knowledge gaps[12] providing the basis for future research. The following research question was formulated: What is known from the literature about pessary self-management for women with POP?

## METHODS AND ANALYSIS
### Types of sources
The scoping review will be conducted using the Joanna Briggs Institute scoping review methodology[13] and reported in accordance with Preferred Reporting Items for Systematic Reviews and Meta-Analyses extension for Scoping Reviews (PRISMA-ScR) guidelines (Supplementary material).[14] As advocated by PRISMA-ScR guidelines, the review has been registered with The Open Science Framework (OSF) (DOI 10.17605/OSF.IO/DNGCP) and the protocol published to ensure transparency and reproducibility.

To be eligible for inclusion in the review, papers must include original research data regarding pessary self-management for women with POP. Pessary self-management and what it entails is poorly defined. However, for the purpose of this scoping review women

who remove or insert their pessary independently will be classed as self-managing. There will be no time limit set as it is acknowledged pessary management of prolapse is a long-standing treatment option and therefore self-management of pessaries may also be. All identified studies will be included regardless of the methodology used to ensure the different aspects and perspectives of pessary self-management for POP will be explored. Non-original articles such as reviews will be excluded to avoid duplication of evidence. However, the reference list of all review articles identified in the search will be checked and all relevant original research cited will be included within this review.

The search aims to include publications from different methodological perspectives and to scope the available evidence related to pessary self-management for POP without presuming what the important outcomes may be. Therefore, rather than the PICO tool which is typically used for systematic reviews of quantitative evidence, the population, concept and context of the research question will be used to structure the search.[15]

### Population
The population in focus are women with POP who use a pessary. It is recognised that the prevalence and severity of POP increases significantly with increasing age.[16] However, women of all ages develop prolapse, with 77% of 18–29-year-old women examined in a routine gynaecological clinic having stage one or two prolapse.[16] Therefore, no limits will be placed on the age of the study population. POP is typically measured using the POPQ system,[17] the measurements of which can be used to determine the stage of prolapse.[17] It is acknowledged the anatomical extent of a prolapse as measured by staging may not correlate with the severity of a woman's symptoms.[18] Therefore, it is advocated women be treated based on the bothersomeness of their symptoms. It can thus be assumed women managing their prolapse symptoms with a pessary were sufficiently bothered to consent to treatment, and the stage of their prolapse is of less significance than this. Therefore, the review will not exclude women based on the stage of their prolapse.

### Concept
The concept to be explored is specifically those who self-manage their pessary for POP. That included studies must relate to this type of pessary is an important distinction, as pessaries can be for either structural or medicinal purposes.[19] Pharmaceutical pessaries are solid tablets containing medicinal products for insertion into the vagina and are typically prescribed when a local effect is desired.[19] While there is limited evidence supporting the use of vaginal oestrogen as a treatment for POP,[20] National Institute for Health and Care Excellence guidelines suggest clinicians consider the use of local oestrogen cream or pessaries for women with symptomatic prolapse and vaginal atrophy.[21] Therefore, there is potential for articles with a focus of pharmaceutical

pessaries for women with POP to be identified during the search. The term mechanical pessaries includes devices designed to manage symptoms of urinary incontinence,[22] with some devices having a dual purpose for management of prolapse and urinary symptoms.[22] Studies will be excluded if the pessaries are solely for urinary incontinence; however, studies with a sample population that includes women using dual purpose pessaries are eligible for inclusion. There is a broad range of pessaries available to manage POP and these are typically classed as either a support or space occupying pessary.[17] While the pessaries have different mechanisms of working, both have the same purpose of reducing the descent of, and symptoms associated with, prolapse. Therefore, studies including both types of pessaries will be included.

## Context

Whether there are differences in the extent to which pessary self-management is offered to, or accepted by women, depending on the country they live in, healthcare provision, or the culture they belong to, is of particular interest to the authors. Therefore, there is no specific context to the question as the authors are interested in evidence related to pessary self-management for POP regardless of the setting where self-management was initiated, or within what country.

Articles will be excluded if they are not accessible in the English language due to feasibility issues, it is acknowledged this may limit the scope and generalisability of findings. Relevant conference abstracts will be included if the full data set has not subsequently been published. While this prevents the authors from appraising the full dataset and detailed description of the study processes, it enables the inclusion of recent research studies which have not yet been published as well as findings which may not be deemed sufficiently significant by the authors to publish.

The search strategy will aim to locate published studies related to pessary self-management. A search of Medline, CINAHL, Embase and PsycInfo will be undertaken to identify articles on the topic. The search terms which will be used for all databases are pessary and self-management or self-care. Searches of all databases will be undertaken between 5 and 7 May 2021. Hand searches will be undertaken throughout May 2021. Data extraction, critical appraisal and synthesis of the results will be undertaken following this.

## Inclusion criteria

- ► Original research.
- ► Pessary for POP.
- ► Published in English language.
- ► Focuses on self-management of pessary for POP.

## Exclusion criteria

- ► Not relevant to subject area.
- ► Not published in English language.
- ► Not original research including case reports.

## Study/source of evidence selection

Following the search, all identified citations will be collated and uploaded and duplicates and non-original research publications removed. The abstracts will then be reviewed for relevance to the review question. In the instance of abstracts which do not explicitly refer to pessary self-management but are relevant to pessaries for prolapse, the full text will be reviewed to check for references to pessary self-management which may not have been deemed as sufficiently significant to include within the abstract. All data identified which is relevant to pessary self-management will be extracted regardless of the overall aim of the publication, the care setting and the level of focus on pessary self-management. A sample of 20% of abstracts will be screened by an independent reviewer to ensure agreement with inclusion or exclusion decision. In the instance of disagreement regarding included or excluded studies not resolved through discussion, a third reviewer will be asked to make the final decision. Potentially relevant sources will be retrieved in full and assessed in detail against the inclusion criteria. Reasons for exclusion of sources of evidence that do not meet the inclusion criteria will be recorded and reported in the scoping review.

## Data extraction

Data will be extracted from papers included in the scoping review using a data extraction tool developed by the reviewers based on scoping review guidance.[13] The extracted data will be entered into the tool electronically via Microsoft Excel. The data extraction form will include details of the author(s), year of publication, population studied, context, methodology used and key findings relevant to pessary self-management. A second reviewer will perform data extraction from a subset of 10% of included articles to ensure a standardised process. If there are discrepancies between both extractions, the potential reasons for this will be explored and the data extraction tool modified accordingly to reduce the likelihood for future discordance. Any amendments to the data extraction tool will be recorded including the reasons for this, for transparency. The final version of the data extraction tool will be included in the scoping review to ensure reproducibility.

## Quality appraisal

Appraising the quality of included studies and the subsequent findings is not typically performed during a scoping review.[15] However, for the purpose of this review, quality will be assessed to determine the weighting of included research findings, as well as identifying potential methodological strengths and limitations of the current evidence base. As identified studies will be included regardless of the methodology used, it is important to identify a quality appraisal tool which can assess qualitative, quantitative and mixed methods research. Therefore, the updated Mixed Methods Appraisal Tool (MMAT) will be used.[23] The MMAT was initially developed in 2006 and

Figure 1

| Studies | Criteria from the Mixed Methods Appraisal Tool | | | | | | | | | | | | | | | | | | | | | | | | |
|---|---|---|---|---|---|---|---|---|---|---|---|---|---|---|---|---|---|---|---|---|---|---|---|---|---|
| | 1.1 | 1.2 | 1.3 | 1.4 | 1.5 | 2.1 | 2.2 | 2.3 | 2.4 | 2.5 | 3.1 | 3.2 | 3.3 | 3.4 | 3.5 | 4.1 | 4.2 | 4.3 | 4.4 | 4.5 | 5.1 | 5.2 | 5.3 | 5.4 | 5.5 |
| Author, date | 0 | 1 | 1 | 1 | 1 | | | | | | | | | | | 1 | 0 | 1 | 1 | 1 | 1 | 1 | 1 | 1 | 1 |
| Author, date | | | | | | | | | | | 0 | 1 | 1 | 1 | 1 | | | | | | | | | | |
| Author, date | | | | | | 1 | 1 | 1 | 0 | 1 | | | | | | | | | | | | | | | |

http://mixedmethodsappraisaltoolpublic.pbworks.com/w/file/fetch/140056890/Reporting%20the%20results%20of%20the%20MMAT.pdf

**Figure 1** Example of presentation of Mixed Methods Appraisal Tool results.

has subsequently been revised to facilitate more efficient quality assessment; therefore, the most recently developed 2018 MMAT will be used for this review.[23] The authors of the revised MMAT discourage reviewers from calculating a single overall score to determine the methodological quality of a paper as it prohibits readers from understanding the specific strengths and limitations of a study[23] Therefore, the quality appraisal of each included study will be presented in a tabular format (figure 1), detailing the rating for each criterion assessed using the MMAT. The same subsample of studies randomly selected by the second reviewer for data extraction quality assurance will be quality appraised by the second reviewer to ensure agreement in the assessment process. The 2018 version of the MMAT has improved content validity compared with previous versions; however, the inter-rater reliability has yet to be established.[24] Therefore, ensuring that there is concordance between both reviewer's assessment of quality will be an important step in the scoping review process.

### Patient and public involvement

Members of the public and pessary users have not directly been involved with development of this protocol or review process. However, the need for research exploring pessary self-management was highlighted by The James Lind Alliance (JLA) Priority Setting Partnership for pessary and prolapse.[25] Several women with experience of pessaries participated in this partnership either as members of the steering group, by attending the consensus workshop or completing questionnaires. Understanding more about self-management was ranked third out of 20 priorities by the JLA Priority Setting Partnership. The topic of the scoping review has therefore previously been identified and prioritised by patients and members of the public.

### Data analysis and presentation

The data will be analysed and presented in numerical and tabular format to describe the current evidence base, for example, the extent of identified literature, the context of included research such as the country of origin and the nature of the research. This will provide a scope of the existing evidence related to pessary self-management as a key objective of this review. Furthermore, the key findings of included research will be analysed for emerging themes. A narrative description of the themes and a summary of the evidence related to this will be presented in text and numerical format as appropriate.

### Ethics and dissemination

No ethical or Health Research Authority approval is required to undertake the scoping review. However, it has been registered with The OSF (DOI 10.17605/OSF.IO/DNGCP) to ensure quality through transparent reporting and to prevent overlapping or duplicate work being undertaken prior to publication of the review findings.[26] The scoping review will be disseminated by publication in a peer reviewed journal and submitted for presentation at a national conference. The findings will also inform subsequent exploratory work regarding pessary self-management.

**Contributors** LD devised the scoping review question, methodology and drafted this manuscript. RK and DD substantively contributed to the development of the scoping review question, methodology and revised and approved this manuscript.

**Funding** Lucy Dwyer, Clinical Doctoral Research Fellow, NIHR300519 is funded by Health Education England (HEE)/National Institute for Health Research (NIHR) for this research project.

**Competing interests** LD and RK are coapplicants of the NIHR/HTA funded Treatment of Prolapse with Self-Care Pessary (TOPSY) study.

**Patient consent for publication** Not applicable.

**Provenance and peer review** Not commissioned; externally peer reviewed.

**ORCID iD**
Lucy Dwyer http://orcid.org/0000-0002-0284-873X

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
