## [Reviewer comments · BMJ Open]

ARTICLE DETAILS

TITLE (PROVISIONAL)	What is known from the existing literature about self-management of pessaries for pelvic organ prolapse? A scoping review protocol.
AUTHORS	Dwyer, Lucy; Dowding, Dawn; Kearney, R

VERSION 1 – REVIEW

REVIEWER	Rantell, Angela King's College Hospital, Department of Urogynaecology
REVIEW RETURNED	06-Aug-2021

GENERAL COMMENTS	Thank you for submitting this interesting and thorough review of the planned protocol for a scoping review. My only comments are in relation to further understanding and describing the patients included in the population. 1. is there a presumption that all data are coming from women who have been sized and fitted for their pessary by an appropriately trained health care professional rather than including information from women who have bought their devices online without a formal diagnosis / assessment?2. are you going to include data from papers comparing pessaries for POP to other interventions eg compared to surgery . PFMT / other adjuvant therapy if self management was offered in these studies?
--

REVIEWER	Lough, Kate Glasgow Caledonian University, Nursing, Midwifery and Allied Health Professions Research Unit
REVIEW RETURNED	19-Sep-2021

GENERAL COMMENTS	This will be an important and pragmatic addition to understanding the information about pessary use for prolapse. My only comment would be to consider how best to present the results to maximise understanding and dissemination, and perhaps to consider what future research is required - but also how the research should be conducted.
---

VERSION 1 – AUTHOR RESPONSE

Comment	Action
Please revise the 'Strengths and limitations' section of your manuscript (after the abstract). This section should contain up to five short bullet points, no longer than one sentence each, that relate specifically to the methods. The aims or anticipated results of the study should not be summarised here.	Actioned as advised
Please include the planned start and end dates for the study in the methods section.	Added further detail as requested
All items from the PRISMA-ScR checklist should be included in your manuscript, and the relevant page number listed in the checklist. Please do not leave blanks, and indicate any items that do not apply to your study design as 'Not Applicable'.	Apologies, it appears the edited version to the PDF had not saved which is why it was blank. I have completed this in the word version and included. Please note, a number of items are not applicable as this is the protocol rather than results.
Please include, as a supplementary file, the precise, full search strategy (or strategies) for all databases, registers and websites, including any filters and limits used.	Submitted with revised article
Is there a presumption that all data are coming from women who have been sized and fitted for their pessary by an appropriately trained health care professional rather than including information from women who have bought their devices online without a formal diagnosis / assessment?	Added clarification to page 7 The results manuscript will provide further details about the type of care setting where pessary care was provided.
Are you going to include data from papers comparing pessaries for POP to other interventions eg compared to surgery . PFMT / other adjuvent therapy if self managment was offered in these studies?	Added clarification to page 7
My only comment would be to consider how best to present the results to maximise understanding and dissemination, and perhaps to consider what future research is required - but also how the research should be conducted.	This will be undertaken in the results manuscript depending upon the findings of the review.